# Automated Kinematic Analysis of Closed-Loop Planar Link Mechanisms

**Tatsuya Yamamoto \*, Nobuyuki Iwatsuki and Ikuma Ikeda**

Department of Mechanical Engineering, School of Engineering, Tokyo Institute of Technology, 2-12-1 Ookayama, Meguro-ku, Tokyo 152-8552, Japan; iwatsuki.n.aa@m.titech.ac.jp (N.I.); ikeda.i.aa@m.titech.ac.jp (I.I.)

**\*** Correspondence: yamamoto.t.bm@m.titech.ac.jp

**Abstract:** The systematic kinematic analysis method for planar link mechanisms based on their unique procedures can clearly show the analysis process. The analysis procedure is expressed by a combination of many kinds of conversion functions proposed as the minimum calculation units for analyzing a part of the mechanism. When it is desired to perform this systematic kinematics analysis for a specific linkage mechanism, expert researchers can accomplish the analysis by searching for the procedure by themselves, however, it is difficult for non-expert users to find the procedure. This paper proposes the automatic procedure extraction algorithm for the systematic kinematic analysis of closed-loop planar link mechanisms. By limiting the types of conversion functions to only geometric calculations that are related to the two-link chain, the analysis procedure can be represented by only one type transformation function, and the procedure extraction algorithm can be described as a algorithm searching computable 2-link chain. The configuration of mechanism is described as the "LJ-matrix", which shows the relationship of connections between links with pairs. The algorithm consists of four sub-processes, namely, "LJ-matrix generator", "Solver process", "Add-link process", and "Over-constraint resolver". Inputting the sketch of the mechanism into the proposed algorithm, it automatically extracts unique analysis procedure and generate a kinematic analysis program as a MATLAB code based on it. Several mechanisms are analyzed as examples to show the usefulness of the proposed method.

**Keywords:** kinematics; mechanism; linkage; analysis procedure; kinematic pair

## 1. Introduction

Motion analysis of mechanical linkage is one of the most essential tasks required when we design products with them. For some famous and popular mechanisms, they have been already accomplished by expert researchers. However, the analysis of many other obscure mechanisms has not been accomplished yet. It is not realistic for the researcher to comprehensively investigate the motion analysis for all of these mechanisms for the user's use. Therefore, if product designers consider the use of several types of mechanisms, they must do their own kinematic analysis. There are some kinematic analysis methods for the closed-loop mechanism, such as solving the closed-loop equation of this mechanism, calculating the posture algebraically using geometry, and the systematic analysis.

In general, the closed-loop equation of closed-loop mechanisms becomes a system of transcendental equations that cannot be solved easily. Wamper [1] and Nielsen and Roth [2] introduced the calculation method based on the Dixon determinant [3] in order to make the problem into an eigenvalue problem and derived the numerical solution. The numerical solutions were obtained by the iterative calculation. However, solutions included the dimensions with the imaginary number or unnecessary solutions, such that the mechanism took impossible configuration. Additionally, such an extensive iterative calculation generally requires enough computer resources and computing time.

The computer aided design software, such as SolidWorks [4], Autodesk Fusion360 [5], and Open Cascade [6], which are often used for developing mechanical products actually perform multi-body kinematic analysis with the iterative calculation. However, because the calculation is just a numerical operation and hidden in the black-box, the user cannot explicitly grasp the analysis process and, unfortunately, the calculation result may converge to a value that indicates the different mechanism posture from user's required.

If the mechanism is a simple single loop chain, the output can be described as a function of input value by geometric analysis algebraically. The books written by JSME [7] and McCarthy [8] show the geometric analysis of four-bar linkage. Because the output is described as the algebraic expression, it is calculated directly, and the iterative calculation is not needed. However, in many cases, the output of a multi-loop linkage cannot be described as an algebraic expression.

Many researchers have worked on displacement analysis for kinematic chains that are classified into the Assur group [9–12]. Assur group is a set of kinematic chains developed by Leonid Assur whose degree-of-freedom is zero. A complex structure can be constructed by extending the assure kinematic chain, in other words, the complex linkage mechanism can be regarded as originating from some assure kinematic chains [13]. Therefore, it is expected that the analysis of the general linkage mechanisms will be possible by achieving the analysis of the kinematic chains belonging to the Assur group. However, it is impossible to analyze all of the innumerable Assur kinematic chains. Besides, it is necessary to figure out which Assur kinematic chain constitutes the link mechanism to be analyzed, and this method requires a huge database of the Assur group.

Funabashi [14] and the books [7,15] introduced the systematic analysis method. On this method, mechanisms are regarded as consisting of several types of basic open chain, and their displacements are calculated by the transformation functions. The transformation function can compute the displacement and the time derivation of each type of basic open chain without iterative calculations. Even for a mechanism with a complex configuration, the displacement of the entire mechanism is finally obtained by dividing it into some basic open chains and calculating them sequentially with transformation functions. Calculations of the chains are performed according to a prepared procedure consisting of the complex combination of transformation functions. Therefore, by referring to the procedure, the status of the mechanism and the process of analysis can be explicitly revealed. However, the analysis procedures differ for each mechanism type, and it is difficult for non-expert engineers to find the procedure.

Müller, Mannheim, Hüsing, and Corves developed the linkage and cam design software, namely Mechanism Developer (MechDev), which supports the machine design engineer during the design process [16]. Its kinematic analysis Plug-Ins is based on the combination of analytical and numerical computation methods. Although this software is a user-friendly design, the detailed procedure of the analysis process is not clear for users.

In this paper, the algorithm that automatically extracts the procedure for the systematic kinematic analysis method for planar linkage mechanisms is proposed. The procedure extraction algorithm is simplified to a simple two-link chain searching algorithm by reducing the types of transformation functions. By describing the configuration of the link mechanism in a new matrix format notation, it makes it possible to automatically search for two-link chains with a computer program. In order to test this algorithm, analysis procedures for two closed-loop plane link mechanisms are extracted. Additionally, it is proposed the way to choose the proper procedure from some available candidates. Our approach realizes automatic procedure extraction that is difficult for the previous method while maintaining the advantage of systematic analysis methods, which is, it can clearly show the process of analysis that cannot be grasped by other software. The motion of a mechanism can be calculated analytically without numerical iterative calculations that are done on other software, except for mechanisms that satisfy the particular condition. This is the expanded version of the conference paper in the 25th Jc-IFToMM Symposium [17]. The main expanded contents are (i) the implemented algorithm of Solver process; (ii) the example of analysis for the mechanism including a prismatic joint;

and, (iii) the way to choose appropriate temporal constraints to be applied to the mechanism so that the analysis will be achieved.

## 2. General Matrix Representation of Linkages

The mechanisms are described as the general form that represents topological configurations of mechanisms. There are several kinds of representation the two-dimensional (2D) graph [18,19], the adjacency matrix [20,21], and so on. In automating the extraction of the analysis procedure, the configuration of the mechanism should be expressed in a matrix for handling on a computer. Additionally, it is necessary to describe which joint is connected to which link, in order to search for a two-link chain described later. Hence, in this paper, the configurations of mechanisms are described as the matrix form, "LJ-matrix", which is based on an incidence matrix that denotes the relationship between links and joints. LJ-matrix is proposed, as follows.

$$M = \{m_{i,j}\} \subset R^{N_J \times N_L} \tag{1}$$

$$m_{i,j} = \begin{cases} 0 & (J_i \text{ not on } L_j) \\ a & (J_i \text{ on } L_j) \end{cases} \tag{2}$$

where $J_i$ is the $i$-th pair and $L_j$ is the $j$-th link for $i = 1, 2, ..., N_J$ and $j = 1, 2, ..., N_L$. $N_J$ and $N_L$ are the numbers of pairs and links. $a$ is defined on Table 1.

**Table 1.** The element $a$ of LJ-matrix is determined depending on types and known/unknown of $J_i$.

| $J_i$ Is | Revolute Pair | Prismatic Pair |
|---|---|---|
| Known | $a = -1$ | $a = -2$ |
| Unknown | $a = 1$ | $a = 2$ |

Known pairs mean fixed pairs or pairs that have been calculated their position already. Unknown pairs mean not calculated pairs.

Afterwards, in order to simplify the discussion regarding the extraction process of the analysis procedure that will be introduced later, the mechanisms are simplified under the following three preprocessing rules. (i) All of th links are represented as binary links or structures composed of them. The multiple-pairs link is rewritten as the truss component consisting of binary links (Figure 1). (ii) The active pair is rewritten as the link whose length is a function of the input of the active pair. This link connects both ends of links connecting to the active pair (Figure 2). (iii) Overlapped multiple pairs are recognized as a single pair.

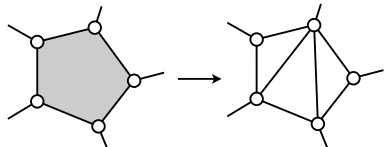

**Figure 1.** A pentagonal link rewritten as a truss consisting of binary links.

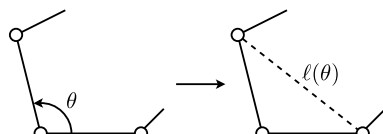

**Figure 2.** An active pair rewritten as an additional binary link in length $\ell(\theta)$.

All the links and pairs of the simplified mechanism are given indices of $L_1, L_2, L_3, \ldots$ and $J_1, J_2, J_3, \ldots$, respectively. In the following discussion, the mechanism to be analyzed will be treated as if it has already been simplified with these rules.

## 3. Procedure Extraction Algorithm

### 3.1. Systematic Analysis Based on Two-Link Chains

Funabashi's systematic analysis method [14] achieves displacement analysis by means of a combination of some simple geometric calculation functions, "Transformation functions". Funabashi proposed seven types of transformation functions. They input the calculated positions of pairs or angles of links and output position of another pair. In this paper, the systematic analysis method is only based on a single transformation function that focuses on two-link chains. A two-links chain is the simplest planar Assur kinematic. For the 0 DoF planar kinematic chains which contain 1 DoF kinematic pairs, the Grübler's equation $F = 3N - 2J - 3 = 0$ can take the smallest number of links $N = 2$ and pairs $J = 3$ that compose a two-links chain. The link mechanism to be analyzed is considered to be regarded as originating from two-links chains belonging to the Assur group. The shape of a two-link chain whose both ends are known is computable and the pair that connects two links of it can be calculated. Figures 3–5 show three types of planar two-link chains treated in this paper. A 3R link chain is a two-link chain consisting of three revolute joints. A 2R1P link chain is a two-link chain that consists of two revolute joints and one prismatic joint. A 1R2P link chain is a two-link chain consisting of one revolute joint and two prismatic joints. In this paper, the middle pair of them is a revolute joint.

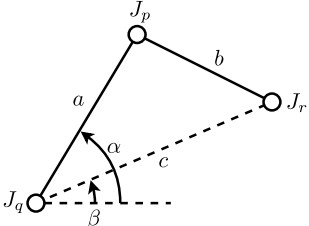

**Figure 3.** 3R link chain.

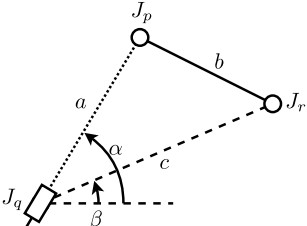

**Figure 4.** 2R1P link chain.

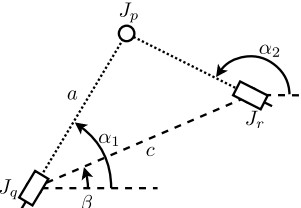

**Figure 5.** 1R2P link chain.

The position and velocity of revolute joint $J_p$ are calculated from the known position of $J_q$ and $J_r$ with dimensions of links, where $\boldsymbol{p}$, $\boldsymbol{q}$, and $\boldsymbol{r}$ are assumed as the position of $J_p$, $J_q$, and $J_r$, respectively.

(i)　3R link chain

$p$ and $\dot{p}$ are calculated from $q$ and $r$ with the link length $a$, $b$, and time derivatives of them.

$$p \;=\; q + \begin{bmatrix} a\cos\alpha \\ a\sin\alpha \end{bmatrix} \tag{3}$$

$$\dot{p} \;=\; \dot{q} + \begin{bmatrix} \dot{a}\cos\alpha - a\dot{\alpha}\sin\alpha \\ \dot{a}\sin\alpha + a\dot{\alpha}\cos\alpha \end{bmatrix} \tag{4}$$

where $\alpha = \beta \pm \arccos\left(\frac{a^2+c^2-b^2}{2ac}\right)$, $\beta = \mathrm{atan2}(\Delta Y, \Delta X)$, $c = ||r-q||$, $[\Delta X,\ \Delta Y]^T = r - q$, $\dot{\alpha} = \dot{\beta} \pm \frac{b\dot{b}-a\dot{a}-c\dot{c}+(\dot{a}c+a\dot{c})\cos\alpha}{ac\sin\alpha}$, $\dot{\beta} = \frac{\Delta\dot{Y}\Delta X-\Delta Y\Delta\dot{X}}{\Delta X^2+\Delta Y^2}$, $\dot{c} = (\dot{r}-\dot{q})^T(r-q)/c$, and $[\Delta\dot{X},\ \Delta\dot{Y}]^T = \dot{r} - \dot{q}$. Each of $p$ and $\dot{p}$ has two solutions. They are symmetric about line connecting $J_q$ and $J_r$ on the $XY$-plane.

(ii)　2R1P link chain

$p$ and $\dot{p}$ are calculated from $q$ and $r$ with the link direction $\alpha$, the link length $b$, and time derivatives of them by Equations (3) and (4), where $a = c\cos\phi \pm \sqrt{b^2 - c^2\sin^2\phi}$, $\phi = \alpha - \beta$, $\dot{a} = \dot{c}\cos\phi - c\dot{\phi}\sin\phi \pm \frac{b\dot{b}-c\dot{c}\sin^2\phi-c^2\dot{\phi}\sin\phi\cos\phi}{\sqrt{b^2-c^2\sin^2\phi}}$, and $\dot{\phi} = \dot{\alpha} - \dot{\beta}$. Each of $p$ and $\dot{p}$ has two solutions. They are symmetric about a line perpendicular to the direction of $J_q$ passing through $J_r$ on the $XY$-plane.

(iii)　1R2P link chain

$p$ and $\dot{p}$ are calculated from $q$ and $r$ with the pair direction $\alpha_1$, $\alpha_2$, and time derivatives of them.

$$p \;=\; q + \begin{bmatrix} a\cos\alpha_1 \\ a\sin\alpha_1 \end{bmatrix} \tag{5}$$

$$\dot{p} \;=\; \dot{q} + \begin{bmatrix} \dot{a}\cos\alpha_1 - a\dot{\alpha}_1\sin\alpha_1 \\ \dot{a}\sin\alpha_1 + a\dot{\alpha}_1\cos\alpha_1 \end{bmatrix} \tag{6}$$

where $a = \frac{\sin\varphi}{\sin\theta}c$, $\theta = \pi - \phi - \varphi$, $\phi = \alpha_1 - \beta$, $\varphi = \pi - \alpha_2 + \beta$, $\dot{a} = \frac{\dot{\varphi}\cos\phi\sin\theta-\dot{\theta}\sin\varphi\cos\theta}{\sin^2\theta}c + \frac{\sin\varphi}{\sin\theta}\dot{c}$, $\dot{\theta} = -\dot{\phi} - \dot{\varphi}$, $\dot{\phi} = \dot{\alpha}_1 - \dot{\beta}$, and $\dot{\varphi} = -\dot{\alpha}_2 + \dot{\beta}$.

By simplifying the mechanism according to preprocessing rules, the position and velocity of every pair are calculated with only transformation function for two-link chains. Figure 6 shows the mechanism whose pair position is calculated by the transformation function for two-link chains. The analysis is achieved with the same procedure, even if the dimensions of each link change.

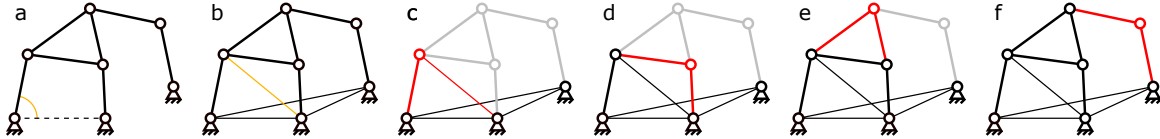

**Figure 6.** The mechanism analyzed by the transformation function for two-link chains: (**a**) The target mechanism (**b**) The active pair is rewritten as a new link. (**c–f**) The position of pair(red) is calculated by the transformation function for two-link chains.

### 3.2. Two-Link Chain Searching Algorithm

On LJ-matrix, computable two-link chains are described, as shown in Figure 7.

If an unknown pair $J_p$ is connected to two known pairs $J_q$ and $J_r$ by links $L_a$ and $L_b$, the $a$-th column and the $b$-th column have a positive element and a negative element. The positive elements are on the $p$-th row and the negative elements are on the $q$-th row or $r$-th row.

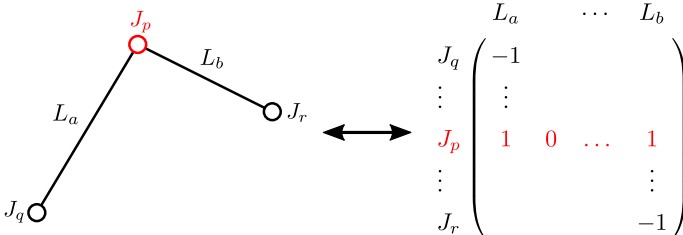

**Figure 7.** A two-link chain described on LJ-matrix.

The algorithm that extracts the analysis order of two-link chains is proposed. Algorithm 1 is the algorithm that searches computable two-link chains and calculates the position and velocity of unknown pairs one by one. This algorithm is named "Solver".

---

**Algorithm 1:** Solver algorithm.

---

LJ-matrix: $M = \{m_{i,j}\}$
while $p \leftarrow$ Repetition of counting up from 1 to $N_J$ (condition: $^\exists m < 0$
　　if $^\exists a, ^\exists b$ such that $m_{p,n} \geq 0 \cap m_{k,n} < 0$ ($\{n,k\} \in \{\{a,q\}, \{b,r\}\}$, $a \neq b$, $q, r = 1, \ldots, N_J$)
　　　% Calculate position and velocity of $p$-th joint with transformation function $f$ %
　　　$\{J_p, \dot{J}_p\} \leftarrow f(J_q, \dot{J}_q, J_r, \dot{J}_r, L_a, \dot{L}_a, L_b, \dot{L}_b)$
　　　$m_{p,all} \leftarrow -m_{p,all}$
　　end
end

---

The Limitation of Solver Algorithm

The Solver algorithm calculates an unknown joint position that is based on the transformation function for the two-link chain. The transformation function is available for computable two-link chains whose both ends are known. Therefore, if a mechanism doesn't have any computable two-link chain during the Solver process, analysis cannot be completed. This means that the linkage mechanism includes Assur kinematic chains other than the two-links chain. In this paper, such a Solver ineffective mechanisms are named "unsolvable mechanism", and others are named "solvable mechanism". Even if a mechanism has the same link chain as a solvable mechanism, it can be an unsolvable mechanism depending on how the active pair is designated. The mechanism presented in Figure 8a is an unsolvable mechanism, as we can recognize that no computable two-link chain exists in Figure 8d.

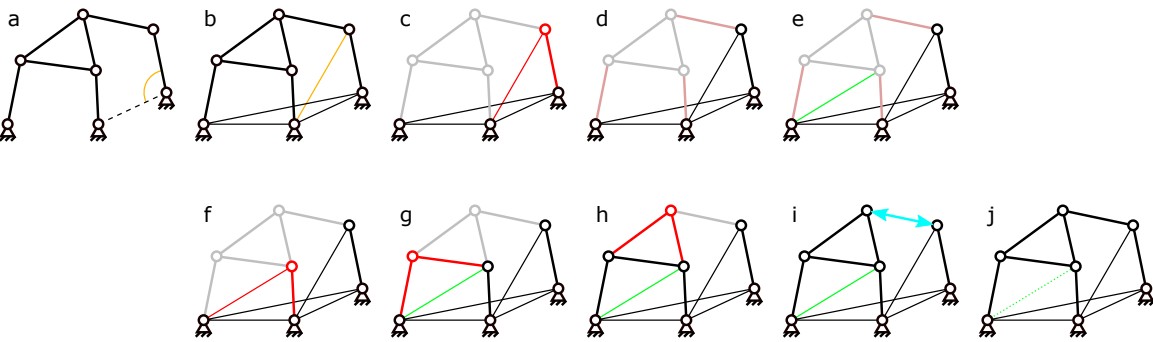

**Figure 8.** Analysis procedure for an unsolvable mechanism: (**a–c**) Same as Figure 6. (**d**) No computable two-link chain exists. (**e**) Add-link(green) is installed. (**f–h**) Same as Figure 6. (**i**) The distance between joints(blue) should be equal to link length. (**j**) Add-link length is optimized.

*3.3. Auxiliary Process for Dealing with an Unsolvable Mechanism*

3.3.1. Additional Temporal Constraint Link

In order to deal with unsolvable mechanisms, additional constraints are temporarily installed into the mechanism. The constraint is described as an additional link that does not exist in the original mechanism (Figure 8e). The additional link makes up a two-link chain, and Solver starts the analysis of mechanism from it (Figure 8f). In this paper, the additional link is named "Add-link". The length of Add-link is temporarily determined to any value. On LJ-matrix, Add-link is described as a new column according to the definitions of LJ-matrix. The requirement of Add-link is that it makes up a computable two-link chain. It means that (i) Add-link connects any known joint and an unknown joint that has been connected to another known joint already and (ii) Add-link does not overlap with existing links. In many cases, there are some Add-link candidates that meet the requirement. Section 5 shows the difference of run time depending on the choices of Add-link candidates.

3.3.2. Resolving Over-Constraint

Add-links gives a new constraint. The degree of freedom is reduced and the mechanism becomes over-constrained. By executing Solver, the positions of each joint are calculated. Because the mechanism contains excess links for installed Add-links, if $n$ Add-links are installed, Solver completes the calculation for all joints without reference to $n$ original links (Figure 8h).

Subsequently, the distance between calculated joints which are on the not referred link is determined independently of link length (Figure 8i). The joints distance depends on the length of Add-links. Therefore, Add-links length has to be optimized, so that the joints distance is equal to the original link length. The optimization of the length of Add-links is realized by the Newton–Raphson method. The evaluation function is given as a system of transcendental equations $f = [f_1, f_2, f_3, \cdots, f_n]^T$, $f_n = d_n - \ell_n$, where $\ell_n$ is the length of the $n$-th not referred to original link by Solver, and $d_n$ is the distance between joints. The new length of Add-link is calculated by iterative calculations of the following equation.

$$\ell_{a_{k+1}} = \ell_{a_k} - J^{-1} f \tag{7}$$

where $\ell_a = [\ell_{a_1}, \ell_{a_2}, \ell_{a_3}, \cdots, \ell_{a_n}]^T$, and $\ell_{a_n}$ is the length of $n$-th Add-link. $J$ is the Jacobian matrix defined as

$$J = \begin{bmatrix} \frac{\partial f_1}{\partial \ell_{a_1}} & \frac{\partial f_1}{\partial \ell_{a_2}} & \cdots & \frac{\partial f_1}{\partial \ell_{a_n}} \\ \frac{\partial f_2}{\partial \ell_{a_1}} & \frac{\partial f_2}{\partial \ell_{a_2}} & \cdots & \frac{\partial f_2}{\partial \ell_{a_n}} \\ \vdots & \vdots & \ddots & \vdots \\ \frac{\partial f_n}{\partial \ell_{a_1}} & \frac{\partial f_n}{\partial \ell_{a_2}} & \cdots & \frac{\partial f_n}{\partial \ell_{a_n}} \end{bmatrix}. \tag{8}$$

$\partial f_n / \partial \ell_{a_n}$ means the infinitesimal change in $f_n$ when the infinitesimal change in the Add-link length is given. It is calculated by the transformation function (Equations (3) and (4) or Equations (5) and (6)).

**4. Flowchart**

Figure 9 shows the flowchart of the proposed algorithm that extracts the systematic kinematic analysis procedure. The algorithm consists of four sub-processes, namely, "LJ-matrix generator", "Solver process", "Add-link process", and "Over-constraint resolver".

Every time the sub-process ends, what the sub-process did is recorded as the analysis procedure logs.

First, the LJ-matrix generator automatically generates the LJ-matrix of the mechanism to be analyzed. Input the schematic diagram of the mechanism drawn by users freely and defining active

pairs and fixed links on the GUI (Figure 10), they are automatically converted into the LJ-matrix by this process. Subsequently, the LJ-matrix generator makes the procedure logs about generated LJ-matrix, fixed links, and active pair.

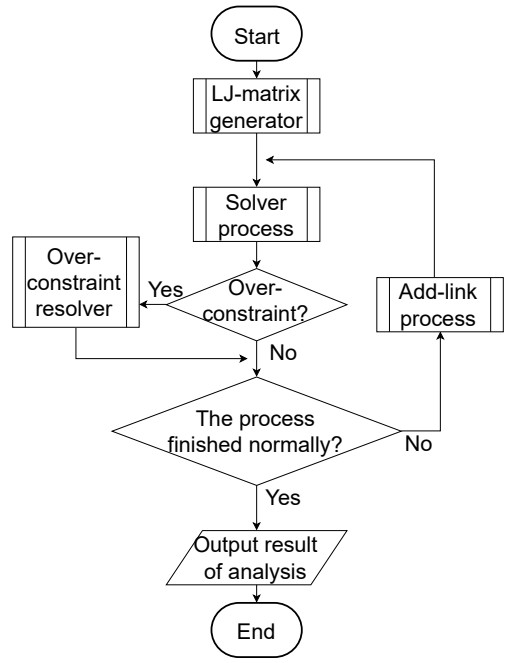

**Figure 9.** Systematic kinematic analysis algorithm for planar linkage.

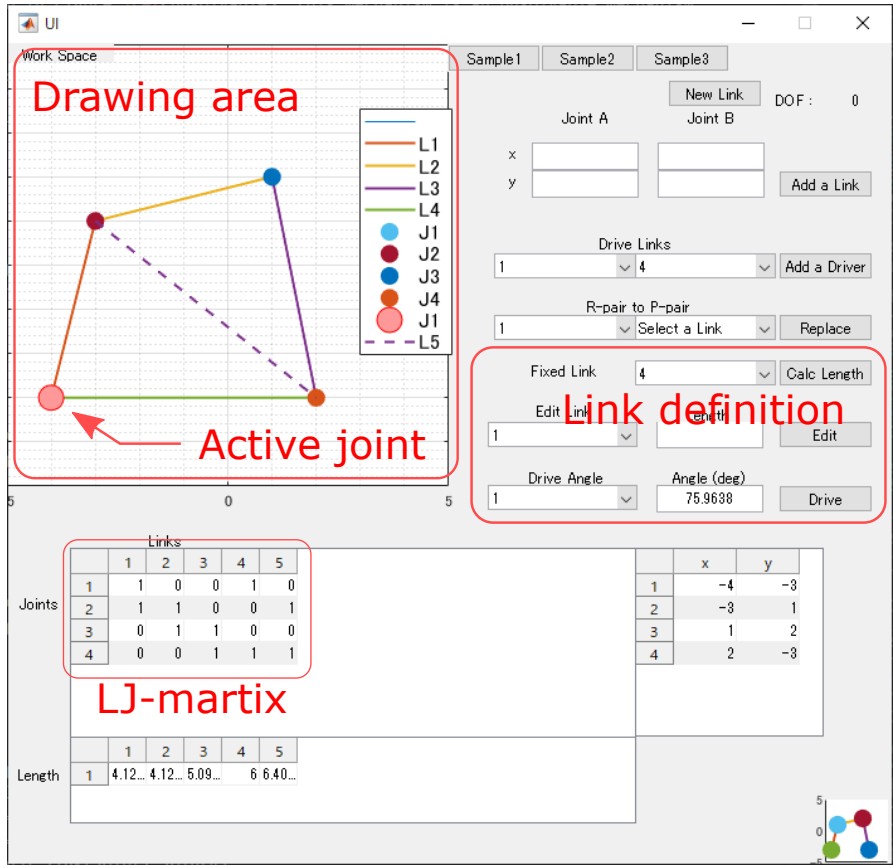

**Figure 10.** GUI for LJ-matrix generator.

The generated LJ-matrix is input into the Solver process, which computes the joint position and velocity with Solver (Section 3.1). In the position calculation, the two-link chains solver (i) or (ii) (Section 3.1) can take two symmetrical position solutions. It is automatically determined according to the input schematic diagram that the solution adopted. Afterwards, the Solver process makes the procedure logs about the target joint to be calculated, dimensions of the two-link chain, and updating LJ-matrix.

In case, the Solver process failed to complete procedure extraction, that is, the target mechanism is unsolvable, Add-link process installs Add-links into the mechanism and its LJ-matrix. Figure 11 shows the flowchart of the Add-link process.

Subsequently, the Add-link process makes the procedure logs about the installed Add-link.

The modified LJ-matrix is input into the Solver process again. If Solver process complete procedure extraction, the Over-constraint resolver re-calculates the length of Add-links based on iterative calculation (Section 3.3.2). Afterwards, the Over-constraint resolver makes the procedure logs about re-calculated Add-links and evaluation function. The convergence of calculations on the Over-constraint resolver means that the kinematic analysis and procedure extraction is completed. After the algorithm is finished, the readable procedure manual and kinematic analysis MATLAB program are generated.

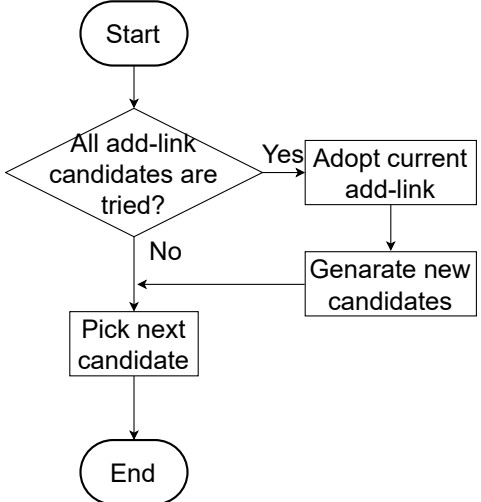

**Figure 11.** Flowchart of Add-link process.

## 5. Results

### 5.1. Problem 1: A Slider-Crank Mechanism

Figure 12a is the schematic diagram of the slider-crank mechanism and Figure 12b is the LJ-matrix of the mechanisms automatically converted from the sketch. $J_2$ is a prismatic joint, and others are revolute joints. On this mechanism, $L_1$ is fixed and the active pair is $J_1$. This mechanism is a solvable mechanism. Listing 1 is the finally generated analysis procedure. By means of the procedure, the motion of the mechanism is automatically drawn when it is given any active pair input by users. Figure 13 shows the motion simulation when the active pair input is linearly increasing.

**Listing 1.** Generated procedure manual.

```
1  lj_mat = fixed_link(lj_mat,link1);
2  joint3 = RRR_links(link2,link5,joint1,joint2);
3  joint4 = PRR_links(link3,link4,joint3,joint2);
```

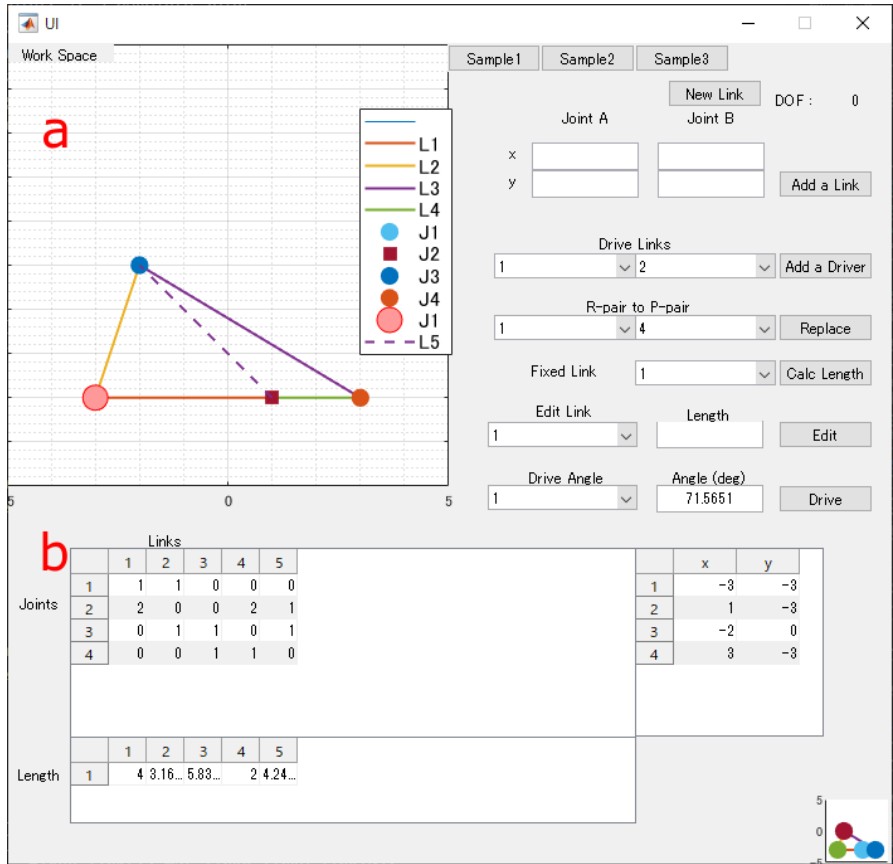

**Figure 12.** The schematic diagram of Slider-crank mechanism (**a**) and generated LJ-matrix (**b**).

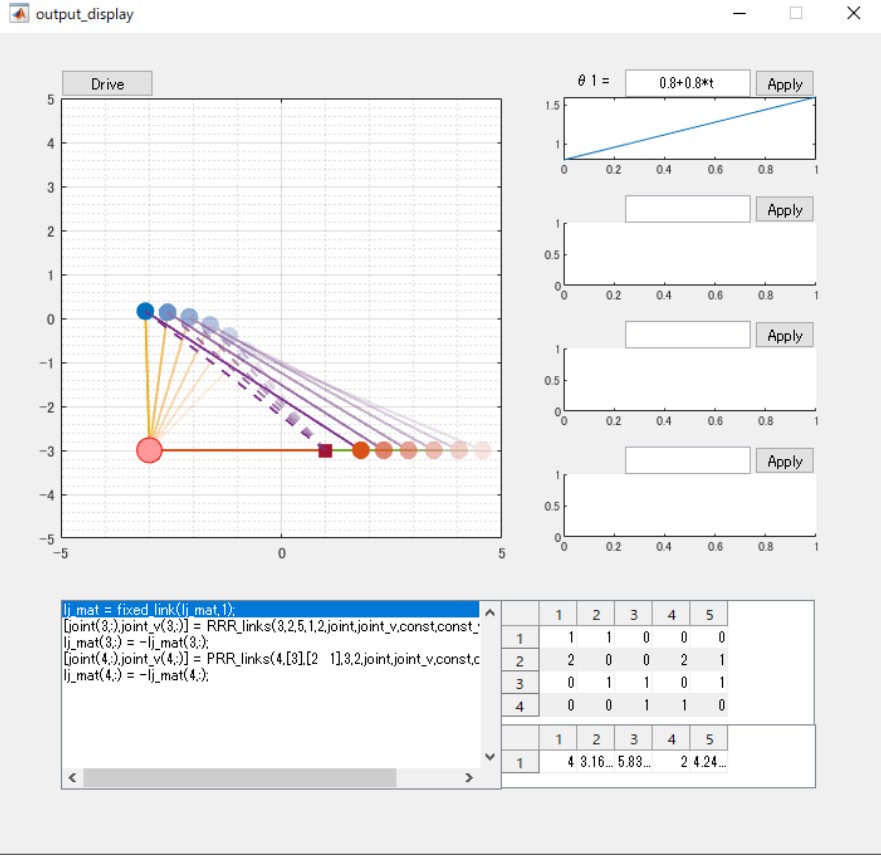

**Figure 13.** The motion simulation of slider-crank mechanism according to the extracted procedure.

## 5.2. Problem 2: A Six-Bar Linkage Mechanism

Figure 14 is the schematic diagram of the six-bar linkage mechanism and the LJ-matrix of the mechanisms automatically converted from the sketch. Every joint is a revolute joint. On this mechanism, the bottom link is fixed and the active pair is $J_7$. This mechanism is an unsolvable mechanism, as shown in Figure 8. In this case, the Add-link was installed into the mechanism as $L_{12}$ connecting $J_1$ and $J_4$ (Figure 14). Listing 2 is the finally generated analysis procedure. By means of the procedure, the motion of the mechanism is automatically drawn when it is given any active pair input by users. Figure 15 shows the motion simulation when the active pair input is linearly increasing.

**Listing 2.** Generated procedure manual.

```
1  lj_mat = fixed_link(lj_mat,link8,link9,link10);
2  joint6 = RRR_links(link7,link11,joint7,joint3);
3  additional_link(joint1,joint4);
4  joint4 = RRR_links(link2,link12,joint3,joint1);
5  joint2 = RRR_links(link1,link3,joint1,joint4);
6  joint5 = RRR_links(link4,link5,joint2,joint4);
7  add_link_optimizer(target_link[6],add-link[12]);
```

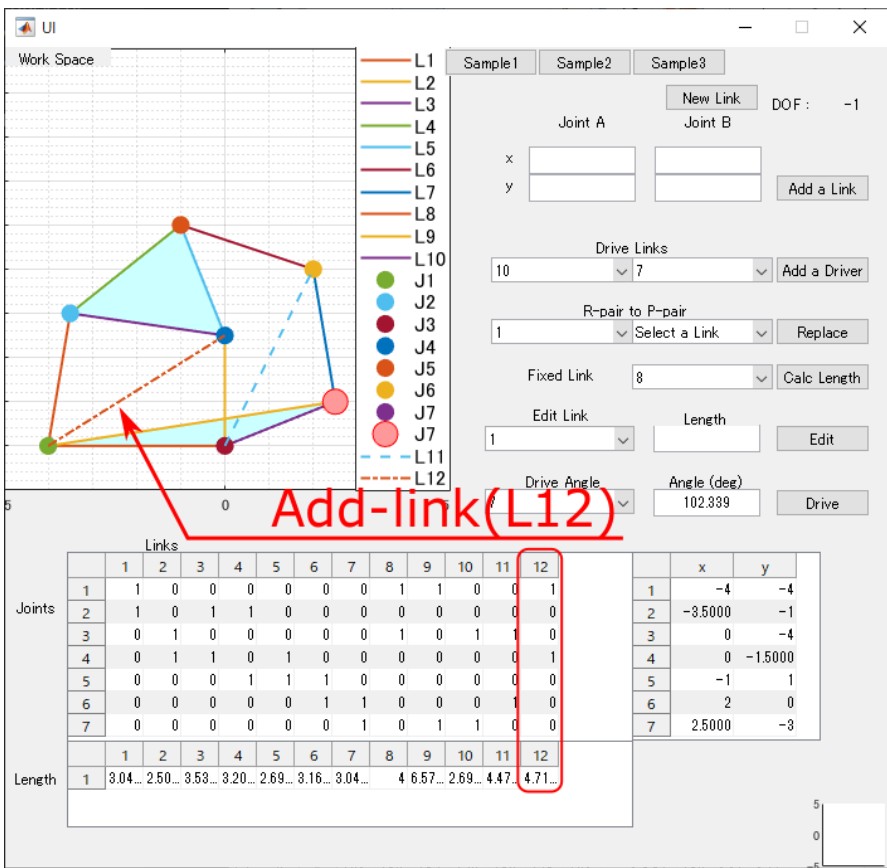

**Figure 14.** 6-bar linkage mechanism drawn on LJ-matrix generator GUI and installed Add-link.

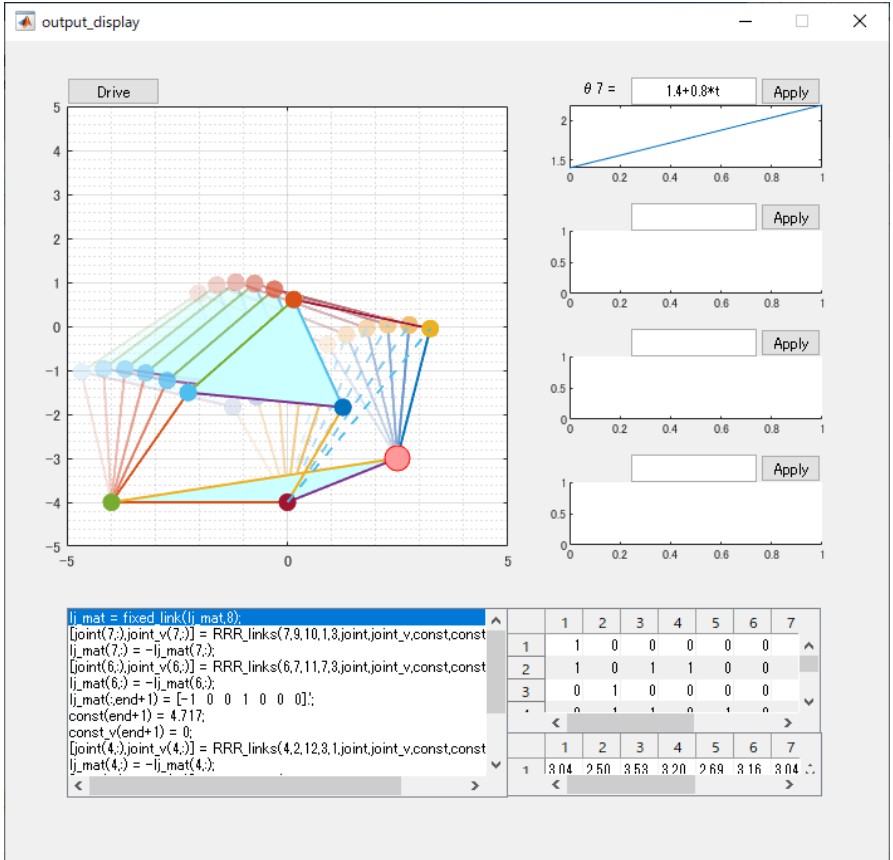

**Figure 15.** The motion simulation of 6-bar linkage mechanism according to the extracted procedure.

## 6. Processing Time of the Algorithm Depending on Add-Link Choice

There are some choices of Add-link that will be installed into the mechanism in order to help the Solver process, as mentioned in Section 3.3.1. Because the proposed analysis algorithm treats installed Add-links like one of the original links in the mechanism, the mechanisms that different Add-links are installed into, but were originally the same link chain are processed as distinctly different mechanisms in the algorithm. Therefore, the configuration of two-link chains and the order of calculations, which are found out by Solver process from each mechanism is diverse. It means that each Add-link candidates finally derive different analysis procedures.

The six-bar linkage mechanism referred in Section 5.2 has nine available candidates for Add-link $L_{12}$, including the adopted one connecting $J_1$ and $J_4$. They satisfy the required condition for Add-link that they configure computable two-link chains with an original link of the mechanism. Table 2 shows all candidates for Add-link $L_{12}$ and connected joints at both ends of them, and they will derive nine types of different analysis procedures.

**Table 2.** Candidates of $L_{12}$ and their connecting joints for 6-bar linkage mechanism.

| $L_{12}(1)$ | $L_{12}(2)$ | $L_{12}(3)$ | $L_{12}(4)$ | $L_{12}(5)$ | $L_{12}(6)$ | $L_{12}(7)$ | $L_{12}(8)$ | $L_{12}(9)$ |
|---|---|---|---|---|---|---|---|---|
| $J_1, J_4$ | $J_1, J_5$ | $J_2, J_3$ | $J_3, J_5$ | $J_2, J_6$ | $J_4, J_6$ | $J_2, J_7$ | $J_4, J_7$ | $J_5, J_7$ |

The best choice is defined as the candidate that minimizes the processing time of the analysis in order to choose one Add-link from the candidates. Of course, the processing time of the analysis algorithm depends on the analysis procedure, and it leads to suitable Add-links.

Figure 16 shows the scatter plot for the processing time of the analysis program. Data of the plot are given by running an actual analysis program for the six-bar linkage mechanism that is shown in

Figure 14, where the active pair angle takes the integer value in the range of −10 degrees to 10 degrees from the initial angle. One hundred times of measurements of processing time are carried out for every input value and Add-link candidates, and the average time of them is calculated as the data for plot. This graph suggests that there is a strong positive correlation between processing time and the number of iterative calculations in the Over-constraint resolver.

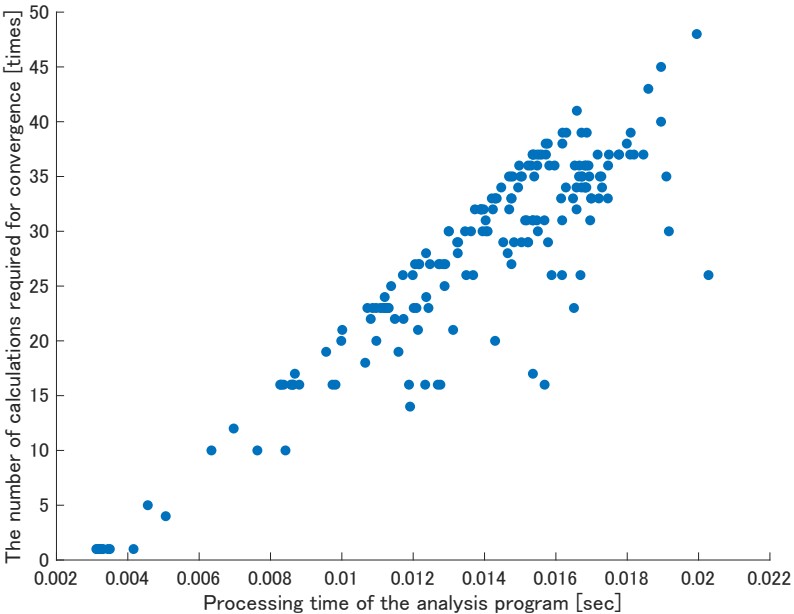

**Figure 16.** The relation between the processing time and the number of iterative calculations in Over-constraint resolver.

Because the number of the calculation is unique for a combination of the choice of Add-link and the active pair angle, the required situation for best Add-link is to minimize the number of iterative calculations. Figure 17 shows the number of iterative calculations and the norm of function values, $f$, of distance between pairs, which is used to obtain the lengths of Add-links mentioned in Section 3.3.1, for each Add-link candidate listed in Table 2. The former is represented as ranges for input displacements −10 degree $< \Delta\theta <$ 10 degree and their average with bars. The latter is represented as "x". In addition, the norm of $f$ is calculated before executing the Over-constraint resolver, and the input displacement is +5 degrees from initial displacement. It is recognized that, around the initial input, these graphs almost resemble, and when an Add-link candidate which makes the norm of $f$ minimum is adopted, the number of iterative calculations also takes the minimum value.

Therefore, in order to choose the best Add-link that minimize the processing time of the analysis procedure, (i) get all candidate that satisfy the required condition for Add-link; (ii) for each candidate, execute Solver process and compute the shape of the mechanism just until running Over-constraint resolver; (iii) calculate the norm of $f$, and find the best candidate that minimizes the norm; and, (iv) adopt this candidate as an Add-link.

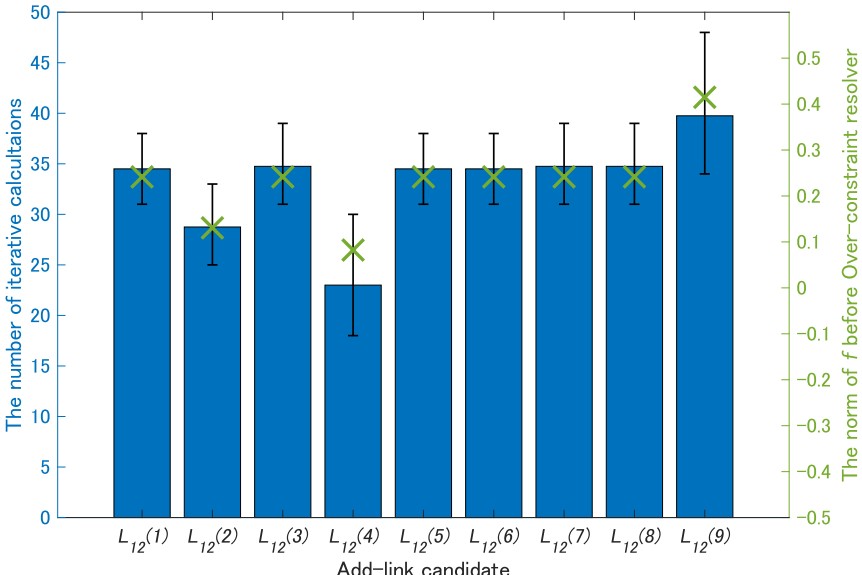

**Figure 17.** Iterative calculations and the norm of function values to obtain lengths of Add-links.

## 7. Conclusions and Future Works

### 7.1. Conclusions

An algoriththat automatically extracts the procedure for the two-link chains based systematic kinematic analysis of planar linkage mechanisms has been proposed. The main results obtained in this paper are summarized, as follows.

(1) LJ-matrix that represents the relationship between links and joints, classification of pairs which are known or unknown has been proposed.
(2) An algorithm for automatically searching two-links chains and extracting the analysis procedure has been established.
(3) This method does not require many kinds of transformation function but is based only on one transformation function.
(4) For a mechanism not analyzed by only the two-links chain based calculations, the analysis is achieved by installing temporally over-constraints as additional links.
(5) The proposed algorithm generates the procedure manual and MATLAB program for the analysis.
(6) The generated procedure manual explicitly reveals the process of analysis.
(7) For several mechanisms, their analysis procedures are extracted and their motions are analyzed with the extracted procedures.
(8) It is shown that the best Add-link candidate that minimizes the processing time of the analysis program is found by assessing the norm of function values to obtain proper Add-links before executing the Over-constraint resolver.

### 7.2. Future Works

We discussed a method for extracting the analysis procedure for kinematics analysis for a general planar linkage mechanism with revolute pairs and prismatic pairs. This method is based on the characteristic of planar linkage mechanisms that the position/orientation of a two-link chain composed of one-degree-of-freedom kinematic pairs and links is determined by the position/direction at both ends. Therefore, even if the linkage mechanism has a planar higher kinematic pair with one degree of freedom such as a plane cam & roller or gear, it is expected that the method introduced in this paper can be used for the analysis of the mechanism. When you add a new type kinematic pair

into this algorithm, the next classification number of the kinematic pair shown in Table 1 is inserted. Besides, it will be required the calculation components of the two-links chain such as Equation (3)–(6) which are based on the kinematic pair combinations shown in Figures 3–5. It is expected that these formulas will become complicated in the case of higher kinematic pairs. In some cases, the iterative calculation may be required in the calculation unit, however, since they are completed in the unit, they do not affect the calculation of the other two-links chains.

In the case of the extension of this method to the spatial linkage mechanism, since the mechanism moves spatially and has multiple DoF kinematic pairs, general two-links chains do not always belong to the Assur group. The Assur group for spatial linkage mechanisms satisfies the Grübler's equation $F = 6(N - J - 1) + \sum_{i=1}^{J} f_i = 0$ and does not include other Assur kinematic chain in itself, where N and J are the numbers of links and pairs and $f_i$ is DoFs of each pair. Therefore, the following problems are considered to exist in order to extend the method proposed in this paper to the spatial linkage mechanism. (i)Classify kinematic chains to Assur groups, (ii) Choose Assur kinematic chains for which can configure easier calculation units, and (iii) Propose a new algorithm to search them from LJ-matrix.

**Author Contributions:** Conceptualization, methodology, software, validation, formal analysis, writing—original draft preparation, writing—review and editing, visualization, T.Y.; supervision, N.I. and I.I.; All authors have read and agreed to the published version of the manuscript.

**Funding:** This research received no external funding.

**Conflicts of Interest:** The authors declare no conflict of interest.

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
