# Peer review of "Automated Kinematic Analysis of Closed-Loop Planar Link Mechanisms"

_machines, doi:10.3390/machines8030041_

Round 1
Reviewer 1 Report
The paper presents a procedure to provide automatically the kinematic analysis of closed-loop planar link mechanisms. The IFToMM conference extension is not sufficiently developed for publication in Machines MDPI journal.
However, there are some concerns to be addressed before publication.
- Introduction section does not provide sufficient state of art background.
- The current paper actually has not explained in detail how proposed method progressed beyond other software.
- The limitation of planar domain is not progressed e.g. 3D spatial analysies.
It is suggested to provide a "Discussion section" for strengthen the innovations factors of proposed algorithm.
- A comparison with commercial sowtare or other existing software could help in evaluating the processing time performance, convergence ability and multi-iteration influence.
Author Response
Thank you for your detailed reviewing.
I will reply to your comments.
>- Introduction section does not provide sufficient state of art background.
I will add statements about the software currently generally used for mechanical design and kinematic analysis in the introduction.
>- The current paper actually has not explained in detail how proposed method progressed beyond other software.
I add statements to the conclusion that the analysis procedure can be specified that cannot be done with existing software.
>- The limitation of planar domain is not progressed e.g. 3D spatial analysies.
It is suggested to provide a "Discussion section" for strengthen the innovations factors of proposed algorithm.
Since this paper deals only with planar mechanisms, no discussion about the spatial mechanism is given.
>- A comparison with commercial sowtare or other existing software could help in evaluating the processing time performance, convergence ability and multi-iteration influence.
Since commercial software is large-scaled and optimized depending on the programming language, I think that comparison of processing time is not appropriate.
Reviewer 2 Report
This paper presents an algorithm to automatically extract the procedure of kinematic analysis of closed loop planar link mechanisms. The algorithm consisted of four processes and demonstrated with MATLAB program.
The paper is very clear and the results will ease the process in analyzing new and complex linkages. The paper fulfils the aim and scope of the Machines. I recommend to open source the MATLAB program to community to boost the complex mechanism analysis in the mechanical engineering community. I my opinion, the paper is accepted after authors go through minor revisions.
I recommend author add explanation of his algorithm in terms of why this one is novel and better than other practices.
Author Response
Thank you for your detailed reviewing.
I will reply to your comments.
> I recommend to open source the MATLAB program to community to boost the complex mechanism analysis in the mechanical engineering community.
Although I did not have any plan to open source the developed MATLAB program to community, I will consider doing so.
> I recommend author add explanation of his algorithm in terms of why this one is novel and better than other practices.
I will introduce the software that are currently used for mechanical design in general, and add statements that the proposed method can specify the analysis procedure that cannot do with existing software into the introduction and conclusion.
Reviewer 3 Report
The paper presents implementation of an algorithm for automatic extraction of a kinetic analysis of the close-loop planar link mechanism. Please find below a list of comments regarding the manuscript:
- Line 6 in the abstract: “The algorithm consists of four sub-processes, namely, "LJ-matrix generator", "Solver process", "Add-link process" and "Over-constraint resolver".” Please rewrite. The Abstract section should highlight what are you most important ideas that you used in designing the new method. We are interested in knowing why these subroutines pay a major role in your method, and not that the algorithm contains a list of subroutines.
- Line 7 in the abstract section: “These processes are programed as a MATLAB program and it can generate an actual MATLAB code to calculate kinematic analysis[1].” Please rephrase. Why is so important that they are written in MATLAB? Why not C++ for example?
- Line 9: “thr” please proofread your manuscript. Such an error in the abstract section is not a good sign.
- Line 12: “This paper is the expended version of the conference paper in the 25th Jc-IFToMM Symposium[1].” It is very important to have sub statement and such paragraph, but placing it as the first paragraph in the introduction is not a good idea. Why should I read the paper after all? Also, “This paper extends our prior work [1].” might provide enough details.
- I would recommend to leave a space between the previous word and the citation.
- The introduction is too short and it must be extended. The literature review must also be extended. The paper starts very abruptly, without providing a proper connection with the state-of-the-art.
- The motivation of this paper is missing.
- Line 51: “Then, the mechanisms are simplified under the following three preprocessing rules.” I the paper proposing these rules or where they proposed in literature? Why was the mechanism simplified?
- Line 77: Please define what is an “2R1P link chain” before using the concept.
- Please plain what is the difference between equations (3) and (5), and between (4) and (6). Please see equation (7).
- Line 118: “The mechanism in Figure 8-a is an unsolvable mechanism.” Please explain why.
- Section 7. Conclusions. “An algorithm for automatically searching two-links chains and extracting the analysis procedure has been established.” What is the novelty of such an algorithm?
Major problems:
- The manuscript must undergo major revision.
- Most of the section are not well written.
- The connection between the section is missing. Not all the concepts used in the manuscript are introduced. The ideas are not liked.
- Th manuscript is written to present a software implementation rather than scientific research.
- There is almost no literature review, no recent papers are cited.
- The manuscript is missing a comparison with the state-of-the-art.
Author Response
Thank you for your detailed review.
I will reply to your comments.
> Line 6 in the abstract: “The algorithm consists of four sub-processes, namely, "LJ-matrix generator", "Solver process", "Add-link process" and "Over-constraint resolver".” Please rewrite. The Abstract section should highlight what are you most important ideas that you used in designing the new method. We are interested in knowing why these subroutines pay a major role in your method, and not that the algorithm contains a list of subroutines.
I will add to the abstract the important ideas I have proposed in this article comparing with the previous work.
> Line 7 in the abstract section: “These processes are programed as a MATLAB program and it can generate an actual MATLAB code to calculate kinematic analysis[1].” Please rephrase. Why is so important that they are written in MATLAB? Why not C++ for example?
The reason why the program is written in MATLAB is that it was easier to implement my idea and develop the GUI compared to other languages such as C ++.
> Line 9: “thr” please proofread your manuscript. Such an error in the abstract section is not a good sign.
I will fix the typographical errors.
> Line 12: “This paper is the expended version of the conference paper in the 25th Jc-IFToMM Symposium[1].” It is very important to have sub statement and such paragraph, but placing it as the first paragraph in the introduction is not a good idea. Why should I read the paper after all? Also, “This paper extends our prior work [1].” might provide enough details. I would recommend to leave a space between the previous word and the citation. The introduction is too short and it must be extended. The literature review must also be extended. The paper starts very abruptly, without providing a proper connection with the state-of-the-art. The motivation of this paper is missing.
I will move the paragraph stating that this article is an extension of an existing article to the end of the introduction.
I will introduced software that is generally used for mechanical design now, and add a statement to the introduction and conclusion that the proposed method can clarify the analysis procedure that existing software cannot.
I insert a blank between the word and the citation.
> Line 51: “Then, the mechanisms are simplified under the following three preprocessing rules.” I the paper proposing these rules or where they proposed in literature? Why was the mechanism simplified?
I will add the description that the mechanism was simplified in order to simplify the discussion later.
>Line 77: Please define what is an “2R1P link chain” before using the concept.
I will add the definition of three kinds of two-link chains including an 2R1P link chain.
> Please plain what is the difference between equations (3) and (5), and between (4) and (6). Please see equation (7).
Equation (5) and (6) will be omitted and Equation (3) and (4) will be referred. Equation (7) will be deleted as a typography error.
>Line 118: “The mechanism in Figure 8-a is an unsolvable mechanism.” Please explain why.
I described that because there is no computable two-link chain in figure 8-d, this mechanism cannot be analized only with the "Solver". I had introduced that we call such the mechanism an "unsolverable mechanism" in the same paragraph.
> Section 7. Conclusions. “An algorithm for automatically searching two-links chains and extracting the analysis procedure has been established.” What is the novelty of such an algorithm?
I will add statements that there is only one transformation function required for analysis compared to the previous research requiring many types of transformation functions to the conclusion.
> Major problems:
> The manuscript must undergo major revision.
> Most of the section are not well written.
> The connection between the section is missing. Not all the concepts used in the manuscript are introduced. The ideas are not liked.
> Th manuscript is written to present a software implementation rather than scientific research.
> There is almost no literature review, no recent papers are cited.
> The manuscript is missing a comparison with the state-of-the-art.
I will revise my manuscript for publication as possible.
Round 2
Reviewer 1 Report
Authors respond to revision comments in inappropriate way.
Please cover major remarks
Author Response
Thank you for your kindly reviewing.
I'm sorry that I could not respond to revision comments in appropriate way.
I will reply to your comments of your previous review report.
I attached the revised manuscript, and each response has corresponding Lines.
Point 1: Introduction section does not provide sufficient state of art background.
Response 1: I have added the new sentences that most of the latest software performs kinematic analysis by hidden numerical calculation process, and the systematic kinematic analysis method dealt with in this research explicitly can show the process of analysis.
Corresponding Lines: Line 33-40
Point 2: The current paper actually has not explained in detail how proposed method progressed beyond other software.
Response 2: I have added the new sentence explaining about following contents.
In previous research, there are many kinds of transformation functions that make up the procedure necessary for systematic analysis, and it is difficult for non-expert users to derive complicated procedures. However, in this research, the procedure can be automatically extracted by re-considering the type of transformation functions.
Corresponding Lines: Line 51-64
Point 3: The limitation of planar domain is not progressed e.g. 3D spatial analysis. It is suggested to provide a "Discussion section" for strengthen the innovations factors of proposed algorithm.
Response 3: Since this paper deals only with planar mechanisms, I guess that discussion about the spatial mechanism is not necessary.
Corresponding Lines: -
Point 4: A comparison with commercial software or other existing software could help in evaluating the processing time performance, convergence ability and multi-iteration influence.
Response 4: Since commercial software is large-scaled and optimized depending on the programming language, I guess that I cannot get results that show only the difference of analysis method clearly even I will make a comparison of each benchmarks.
Corresponding Lines: -

Reviewer 3 Report
-
- The authors are not allowed to cite papers in the abstract section.
- Lines 2-3: “The procedures have been found out by expert researchers’ exhaustive efforts one by one.” Please rephrase.
- Lines 3-4: “This paper proposes the algorithm to automatically extract the procedure of kinematic analysis of […]”. Which algorithm? Please rephrase.
- The abstract section must be improved. An Extensive editing of English language and style required.
- In the introduction section, the first sentence is written twice.
- The first paragraph of the introduction must be rewritten, the authors need to provide a proper introduction of their work. Provide a connection from a general problem to the problem solved in this manuscript. Moreover, provide a motivation of their work and how this proposed method connects with the state-of-the-art methods.
- The literature review must be extended.
- The last paragraph in the introduction section: the first and second sentence start with “this paper”. Moreover, please introduce a connection between the introduction section and the rest of the paper, i.e., the paper outline.
- Section 2: Please introduce a motivation for why such approach was selected.
- Line 67: “(i)All links are to be binary links.” Please rephrase.
- Section 2 ends abruptly, some explanations are missing.
- Section 3. How is this section connected with the previous sections?
- I suggest to use references to the equations or algorithms previously introduced.
- Why was section 3.3 introduced?
- Line 168: Please avoid such 1-line paragraph.
- Line 170: “LJ-matrix generator automatically generates the LJ-matrix of the mechanism to be analyzed, which is input as rough sketch of links and joints of the mechanism.” Please rephrase.
- I recommend to rearrange section 4 as a step-wise algorithm. The whole section contains short sentences with a bad connection between the paragraphs.
- Section 5: Please change the title.
- Figure 12: The notations a and b the figure are unprofessional. The authors need to introduce in the caption an explanation regarding the two parts.
- The sequence of figures 14 and 15 is also unprofessional. The manuscript is not a slide presentation.
- The paper is missing a comparison with state-of-the-art methods or at least other existing software programs.
As a general comment, the quality of the manuscript must be further improved.
Please note that these comments are presented to the authors with the goal of improving the manuscript. The authors must modify the manuscript or provide a good argumentation on why the comment was not complemented by a modification in the manuscript.
I recommend the authors that when making a revision to provide also a version of the manuscript where the introduced modifications are clearly visible to help the reviewers.
Author Response
Thank you for your kindly reviewing.
I will reply to your comments.
I attached the revised manuscript, and each response has corresponding Lines.
Please read my response while referring to the manuscript.
Point 1: The authors are not allowed to cite papers in the abstract section.
Response 1: I removed citing papers in the abstract.
Corresponding Lines: Line 1
Point 2: Lines 2-3: “The procedures have been found out by expert researchers’ exhaustive efforts one by one.” Please rephrase.
Response 2: I rephrased the sentence as following.
... , expert researchers can accomplish the analysis by searching for the procedure by themselves, ...
Corresponding Lines: Line 5-6
Point 3:Lines 3-4: “This paper proposes the algorithm to automatically extract the procedure of kinematic analysis of […]”. Which algorithm? Please rephrase.
Response 3: I rephrased the sentence as following.
This paper proposes the automatic procedure extraction algorithm for the systematic kinematic analysis of closed-loop planar link mechanisms.
Corresponding Lines: Line 7-8
Point 4: The abstract section must be improved. An Extensive editing of English language and style required.
Response 4: I rewritten the abstract section.
Corresponding Lines: Line 2-11
Point 5: In the introduction section, the first sentence is written twice.
Response 5: I removed this error.
Corresponding Lines: -
Point 6: The first paragraph of the introduction must be rewritten, the authors need to provide a proper introduction of their work. Provide a connection from a general problem to the problem solved in this manuscript. Moreover, provide a motivation of their work and how this proposed method connects with the state-of-the-art methods.
Response 6: I rewritten the introduction section.
Corresponding Lines: Line 20-27
Point 7: The literature review must be extended
Response 7: I could not make appropriate revision about this point
Corresponding Lines: -
Point 8: The last paragraph in the introduction section: the first and second sentence start with “this paper”. Moreover, please introduce a connection between the introduction section and the rest of the paper, i.e., the paper outline.
Response 8: I rewritten the introduction section.
Corresponding Lines: Line 49-65
Point 9: Section 2: Please introduce a motivation for why such approach was selected.
Response 9: I added the sentences the reason a motivation for why such approach was selected.
Corresponding Lines: Line 72-74
Point 10: Line 67: “(i)All links are to be binary links.” Please rephrase
Response 10: I rephrased the sentence as following.
All links are represented as binary links or structures composed of them.
Corresponding Lines: Line 83
Point 11: Section 2 ends abruptly, some explanations are missing.
Point 12: Section 3. How is this section connected with the previous sections?
Response 12, 13: I added the sentences connecting these sections.
Corresponding Lines: Line 88-89
Point 13: I suggest to use references to the equations or algorithms previously introduced.
Response 13: I could not find the equations or algorithms that should be use references.
Corresponding Lines: -
Point 14: Why was section 3.3 introduced?
Response 14: The reason why was section 3.3 introduced is to refer the mechanisms that cannot analyzed only with Solver process and the way to deal with such mechanisms. In order to clarify the motivation, I reconsidered the section numbers and their titles.
Corresponding Lines: Line 145 and Line 154
Point 15: Line 168: Please avoid such 1-line paragraph.
Response 15: I rebuild the paragraph of this section.
Corresponding Lines: Line 206
Point 16: Line 170: “LJ-matrix generator automatically generates the LJ-matrix of the mechanism to be analyzed, which is input as rough sketch of links and joints of the mechanism.” Please rephrase.
Response 16: I rephrased the sentence as following.
First, LJ-matrix generator automatically generates the LJ-matrix of the mechanism to be analyzed. Input the schematic diagram of the mechanism drawn by users freely and defining active pairs and fixed links on the GUI (Figure 10), they are automatically converted into the LJ-matrix by this process.
Corresponding Lines: Line 187-189
Point 17: I recommend to rearrange section 4 as a step-wise algorithm. The whole section contains short sentences with a bad connection between the paragraphs.
Response 17: I rebuild the paragraph of this section, and added the sentences connecting these paragraphs.
Corresponding Lines: Line 187-206
Point 18: Section 5: Please change the title.
Response 18: I changed the title to "Result".
Corresponding Lines: Line 207
Point 19: Figure 12: The notations a and b the figure are unprofessional. The authors need to introduce in the caption an explanation regarding the two parts.
Response 19: I changed the caption to explain the two parts.
Corresponding Lines: - (The caption of Figure 12.)
Point 20: The sequence of figures 14 and 15 is also unprofessional. The manuscript is not a slide presentation.
Response 20: I unified them as figure 15 and changed references in the sentences.
Corresponding Lines: -
Point 21: The paper is missing a comparison with state-of-the-art methods or at least other existing software programs.
Response 21: Since existing software is large-scaled and optimized depending on the programming language, I guess that I cannot get results that show only the difference of analysis method clearly even I will make a comparison of processing time performance.

This manuscript is a resubmission of an earlier submission. The following is a list of the peer review reports and author responses from that submission.